# Targeted Toxins for the Treatment of Prostate Cancer

**DOI:** 10.3390/biomedicines9080986

**Published:** 2021-08-09

**Authors:** Philipp Wolf

**Affiliations:** 1Department of Urology, Medical Center, University of Freiburg, 79106 Freiburg, Germany; philipp.wolf@uniklinik-freiburg.de; Tel.: +49-761-270-28921; 2Faculty of Medicine, University of Freiburg, 79106 Freiburg, Germany

**Keywords:** prostate cancer, targeted toxins, targeting, immunogenicity, intracellular trafficking, resistance

## Abstract

Prostate cancer is the second most common cancer and the fifth leading cause of cancer deaths worldwide. Despite improvements in diagnosis and treatment, new treatment options are urgently needed for advanced stages of the disease. Targeted toxins are chemical conjugates or fully recombinant proteins consisting of a binding domain directed against a target antigen on the surface of cancer cells and a toxin domain, which is transported into the cell for the induction of apoptosis. In the last decades, targeted toxins against prostate cancer have been developed. Several challenges, however, became apparent that prevented their direct clinical use. They comprise immunogenicity, low target antigen binding, endosomal entrapment, and lysosomal/proteasomal degradation of the targeted toxins. Moreover, their efficacy is impaired by prostate tumors, which are marked by a dense microenvironment, low target antigen expression, and apoptosis resistance. In this review, current findings in the development of targeted toxins against prostate cancer in view of effective targeting, reduction of immunogenicity, improvement of intracellular trafficking, and overcoming apoptosis resistance are discussed. There are promising approaches that should lead to the clinical use of targeted toxins as therapeutic alternatives for advanced prostate cancer in the future.

## 1. Prostate Cancer

The fight against prostate cancer (PC) is a major challenge. Improvements in diagnosis and treatment in recent years have led to reduced or at least stable mortality rates in most countries [1]. Nevertheless, with almost 1.4 million estimated new cases and 375,000 deaths PC remains the second most frequent cancer and the fifth leading cause of cancer deaths in men worldwide [2]. Primary tumors can be successfully treated by surgery or radiation. Curative treatment, however, is no longer possible in advanced stages and many patients develop castration resistant PC (CRPC) after androgen deprivation therapy. The median survival of patients with metastatic CRPC ranges between 15 and 36 months [3,4]. Therefore, new treatment options are urgently needed for advanced stages of the disease.

## 2. Targeted Toxins

Recognizing that naturally occurring bacterial, plant or human toxins can effectively kill human cells, including tumor cells, targeted toxins have been developed as therapeutic alternatives for numerous tumor entities [5]. Targeted toxins are chemical conjugates or fully recombinant proteins, which consist of two main parts: a binding domain and a toxin domain. The binding domain ensures specific targeting of antigens on the surface of tumor cells. When the binding domain is an antibody, antibody fragment, or cytokine, targeted toxins are also called immunotoxins. When the binding domain is a ligand, growth factor, or hormone, they can also refer to as fusion (protein) toxins. Ribosome-inactivating proteins (RIPs) are preferred toxins for the generation of targeted toxins. They come from bacteria or plants, have enzymatic activity, and irreversibly inhibit protein biosynthesis in eukaryotic cells [6]. Bacterial RIPs have evolved under evolutionary pressure as virulence factors to effectively damage host cells. Prominent examples comprise, e.g., *Diphteria* toxin (DT) from *Corynebacterium diphtheriae* [7] or *Pseudomonas* Exotoxin A (PE, ETA) from *Pseudomonas aeruginosa* [8]. Plant toxins evolved to defend against pests and comprise Ricin A from *Ricinus communis* or Saporin from *Saponaria officinalis* [9].

Some RIPs, like DT, PE, or Ricin A, naturally contain binding domains against cell surface structures that are widely distributed on eukaryotic cells [10]. This creates a broad spectrum of host cells that can be attacked by the toxins. Exchange of the natural binding domains against antibodies or ligands are therefore done in order to achieve specific binding to target antigens on the surface of cancer cells [11]. Some RIPs also comprise sequences for the toxin translocation via eukaryotic membranes that can be used for the generation of targeted toxins for guidance of the toxin domain into the cytosol [6].

In general, targeted toxins exert their cytotoxicity as shown in Figure 1. After systemic application they are transported via the bloodstream to the tumor and bind with help of their binding domain to antigens on the surface of the tumor cells [12]. Then the targeted toxin/antigen complex is taken up into the cells by receptor-mediated endocytosis. For DT-based targeted toxins, the acidic pH in the endosomes causes conformational changes of the translocation domain, resulting in a large channel that allows release of the toxin domain into the cytoplasm [7]. The translocation domain of PE undergoes a conformational change, which makes a furin-cleavable motif accessible. The protease furin cleaves the toxic domain from the binding domain. After cleavage the binding and toxin domains are still connected by a disulfide bond, which encompasses the furin cleavage site. There is evidence that there is an unfolding event, possibly under the influence of chaperones, which leads to a surface exposure of the disulfide bond, which is finally reduced, presumably by protein-disulfide-isomerases [8,13]. PE and Ricin A are transported via the *trans-*Golgi network retrogradely into the Endoplasmatic Reticulum (ER). There, they use the ER-associated protein degradation (ERAD) pathway for the transport of misfolded cellular proteins for ubiquitination and subsequent proteasomal degradation to enter the cytosol [14]. Once there, the toxins inhibit the cellular protein biosynthesis by ADP-ribosylation of the eukaryotic elongation factor 2 (eEF-2) on the ribosomes (PE, DT) [8,15], or by rRNA N-glycosidase activity to depurinate a critical adenine, which results in the inability of the ribosome to bind to eEF-2 (Saporin, Ricin A) [16,17]. The inhibition of protein biosynthesis results in an induction of the intrinsic apoptotic pathway and finally death of the tumor cells. 

Molecular mechanisms that lead to the induction of apoptosis after inhibition of protein biosynthesis are not yet fully understood and appear to be toxin- and cell-dependent. In PC cells, PE-based targeted toxins have been shown to degrade anti-apoptotic Bcl-2 proteins, specifically Mcl-1 and Bcl-2A1 [18,19]. This results in an activation of the pro-apoptotic effectors Bak and Bax followed by cytochrome C release from the mitochondria and induction of apoptosis (Figure 2). In addition, protein biosynthesis inhibition can directly induce mitochondrial stress, lipid oxidation, DNA damage, and thus apoptosis. In addition, MAPK/JNK activation might be involved, leading to an upregulation of pro-apoptotic effector proteins, downregulation of anti-apoptotic proteins, and triggering of mitochondrial stress (rev. in [20]).

## 3. Targeted Toxins in the Clinic

Three targeted toxins were approved by the FDA for cancer treatment until today: Denileukin diftitox (Ontak^®^), consisting of IL2 as binding and DT as toxin domain, for the treatment of cutaneous T cell lymphoma (CTCL) in 1999 [21], Moxetumomab pasudotox (Lumoxiti^®^), consisting of an anti-CD22 antibody fragment and PE38 from *Pseudomonas* Exotoxin A, for the treatment of hairy cell leukemia (HCL) [22], and Tagraxofusp-erzs (Elzonris^®^), consisting of IL3 and DT, for the treatment of blastic plasmacytoid dentritic cell neoplasm (BPDCN) [23], both in 2018. Various targeted toxins against solid tumors are currently tested in clinical trials, but have not been approved yet (ref. in [24]).

In summary, targeted toxins are very potent drugs for cancer therapy that specifically target cancer cells and have a high antitumor activity. Constitutive internalization of targeted toxins after antigen binding results in an intracellular enrichment. Moreover, due to their enzymatic activity targeted toxins elicit a much higher cytotoxicity than antibodies or inhibitors, which only show a stoichiometric one to one binding to their targets and act by blocking of signaling pathways in the target cells. 

Our review focuses on the preclinical development and optimization of targeted toxins for the treatment of PC. For this, PubMed database was used to identify studies on targeted toxins against prostate cancer (main keywords: prostate (cancer), (targeted) toxins, immunotoxins). 

## 4. Targeted Toxins against Prostate Cancer

Generally, PC is regarded as a suitable target for targeted toxin therapy, because (a) PC cells are generally slowly growing and express well described target antigens, (b) PC metastases predominantly involve lymph nodes and bones, locations that are readily accessible to circulating targeted toxins, and (c) the prostate-specific antigen (PSA) serum marker is established for the early detection of metastases and for monitoring the therapeutic efficacy. 

In the last years, different targeted toxins were generated for the treatment of PC and were successfully tested in view of binding, internalization and cytotoxicity on PC cells representing different advanced stages of the disease (Table 1). Different antigens were targeted that are present as transmembrane proteins on the PC cell surface, not shed into circulation and internalized after targeted toxin binding. The most prominent examples are the epidermal growth factor receptor (EGFR) and the prostate specific membrane antigen (PSMA). Both show enhanced expression in advanced metastatic and castration-resistant stages of PC and are correlated with worse prognosis and poor clinical outcome [25,26,27,28]. 

Targeted toxins have not yet been used in clinical trials against PC, because the treatment of solid tumors provides challenges that are not easy to solve. The challenges comprise immunogenicity, low antigen binding, endosomal entrapment, and lysosomal/proteasomal degradation of the targeted toxins. Moreover, tumor cells might reduce their efficacy by formation of a dense tumor microenvironment, low target antigen expression, and apoptosis resistance (Figure 1).

In the following sections solutions for overcoming these challenges in view of effective targeting, reduction of immunogenicity, improvement of intracellular trafficking, and overcoming resistance are discussed (Table 2). 

## 5. Effective Targeting

### 5.1. Challenges

Major obstacles for the treatment of prostate tumors with targeted toxins are an elevated tumor interstitial fluid pressure and a TME, which is marked by stromal cells, extracellular matrix, defective blood, and lymphatic vessels [57]. They prevent an effective extravasation and tumor penetration of macromolecules like targeted toxins following systemic application [58]. 

### 5.2. Solutions for Effective Targeting

#### 5.2.1. Surmounting the TME

To surmount the TME, a focal targeted toxin treatment of local prostate tumors may be considered. In a study of Husain and colleagues repeated intratumoral injections of the targeted toxin IL4-Ctx consisting of the IL4 ligand and the PE variant PE38KDEL as toxin domain into s.c. growing LNCaP or DU145 tumors in nude mice was found to be more effective compared to intraperitoneal or intravenous injections. Complete remissions in all animals bearing DU145 tumors were recorded after treatment with high doses of 500 µg/kg IL4-Ctx [39]. In another study, growth of PC-3 tumors expressing IL13 was significantly inhibited by intratumoral injection of the targeted toxin IL13-PE [36]. The effective treatment of PC by intratumoral application of toxin conjugates was confirmed in a recent study. Rogers and colleagues used urea-based small PSMA inhibitors for the targeted delivery of the PE35 toxin domain into PC cells. Intratumoral application of the conjugate into mice bearing PSMA expressing PC-3 tumors resulted in a >50% average reduction in tumor size after two weeks and a >90% reduction in PSA levels in animals with LNCaP xenografts [59]. Like other focal treatment modalities of PC, such as high intensity focal ultrasound, focal laser ablation, photodynamic therapy, or focal cryotherapy, focal targeted toxin treatment could be considered suitable for patients placed between active surveillance and whole-gland extirpative therapy [60]. Independent from prostate volume, a PSA value of <15 ng/mL, clinical stage T1c-T2a, Gleason score 3 + 3 or 3 + 4, and a life expectancy of >10 years are recommended as inclusion criteria for a focal therapy [61]. Stereotactic injection of the targeted toxins would allow to increase intratumoral availability and systemic toxicities could be diminished or avoided. Moreover, since focal therapy would require considerably fewer targeted toxin doses than systemic therapy, costs for targeted toxin manufacturing could be reduced. It must be taken, however, into account that prostate tumors are often multifocal and treatment of only the index lesions could not be sufficient to control the disease for a long time [62].

Pretreatment of prostate tumors with drugs that damage solid tumor masses and the TME could help systemic applicated targeted toxins to reach the tumor cells better (rev. in [63]). For example, taxanes were found to decompress blood vessels, lower interstitial fluid pressure and reduce the density of different solid tumors [64,65,66]. We found a 7.5- to 19-fold increased cytotoxicity of the anti-PSMA targeted toxin D7(VL-VH)-PE40 after combination with docetaxel on LNCaP and C4-2 cells. Moreover, an enhanced in vivo antitumor activity of the targeted toxin was demonstrated after pretreatment with docetaxel in a PC SCID mouse xenograft model [46]. Further studies must show whether the high efficacy of the combination therapy was due to a pre-damage of the tumors by chemotherapy and due to increased accessibility of the tumor cells for the targeted toxin.

#### 5.2.2. Enhancing Tumor Penetration and Affinity

Further strategies to enhance the efficacy of targeted toxins against solid tumors are to reduce their size to facilitate tumor penetration and to optimize their binding affinity. For first generation targeted toxins against PC, full-length IgG molecules, like anti-CD44, anti-UB domain-containing protein 1 (CDCP1), or anti-EGFR mAbs were used [29,30,33]. Macromolecules like IgG with about 150 kDa in weight, however, can only penetrate tumor masses poorly and can be taken up by Fc receptor expressing cells [67,68]. This overall leads to a reduced number of therapeutically active targeted toxins on the tumor side. For enhanced tumor penetration, targeted toxins of newer generations are therefore preferably constructed using antibody fragments, such as single-chain variable fragments (scFv) or single-domain antibodies (sdAb) with a molecular mass of only about 30 or 12–15 kDa, respectively [48,69].

A reduction in size for an enhanced tumor penetration can also be achieved by deleting parts of the toxin domains. For example, we generated the anti-PSMA targeted toxin hD7-1(VL-VH)-PE24 containing a truncated PE24 toxin domain with 24 kDa in size and the targeted toxin hD7-1(VL-VH)-PE40 with the 40 kDa toxin domain as counterpart. With hD7-1(VL-VH)-PE24 IC_50_ values of 82 pM and 24 pM on LNCaP and C4-2 cells, respectively, were reached after 72 h incubation. For hD7-1(VL-VH)-PE40 similar IC_50_ values of 25 pM and 6 pM were determined [44]. This example proves that size reduction of a toxin domain can be done without or only marginally impairment of cytotoxicity. Further experiments will show, if the PE24 based targeted toxin shows enhanced tumor penetration in vivo.

A reduction in size, however, can also lead to a significant change in the pharmacokinetics of a targeted toxin. A molecular weight of <60 kDa can result in a fast renal clearance and a reduced serum half-life and consequently requires repeated or continuous systemic administration in the patient. The attachment of polyethylene glycol (PEGylation) is one method to increase the molecular weight and the plasma half-life of targeted toxins [70].

ScFv have generally lower affinity to the target antigen in comparison to their parental full length IgG due to monovalent binding and absence of stabilizing constant regions [71,72]. ScFv with affinities in the low nM range are generally discussed most suitable for the construction of targeted toxins, because scFv with lower affinity failed to significantly accumulate in tumors, whereas scFv with higher affinities showed a limited tumor penetration [73,74]. Changing the order of the variable domains of the heavy (VH) and light chain (VL) of a scFv in a targeted toxin can also lead to changes in binding affinity. We constructed two targeted toxins against PC consisting of the anti-PSMA scFv D7 as binding domain and PE40 as toxin domain with different VH-VL orientations. Whereas for the targeted toxin with VH-VL orientation a K_d_ value of 57.7 nM was calculated on C4-2 cells, its counterpart with the scFv in VL-VH orientation showed a an about 3-fold enhanced affinity (K_d_ = 18.3 nM) [49].

#### 5.2.3. Enhancing Target Antigen Expression

Generally, targeted toxins show higher cytotoxicity in cells with a higher target antigen expression than with a lower one [32,35]. Therefore, increasing the antigen expression in PC cells was examined as a strategy to enhance the sensitivity of targeted toxins [37]. In a study of Gonzalez-Moreno and colleagues, PC-3 cells were transfected with the gene of the peptide adrenomedullin (AM). An upregulation of about 100 genes was found after transfection involved in regulating cell cycle arrest, apoptosis, cytoskeleton, cell adhesion, extracellular matrix, immune function and transcription, including the IL-13 receptor subunit a2 (IL-13Ra2) [75]. Treatment of PC-3 cells with the AM peptide also led to an enhanced expression of IL-13Ra2 protein. After five days of incubation of the AM transfected PC-3 cells with the targeted toxin IL13-PE, consisting of IL13 and the PE38 toxin domain, an IC_50_ value of 5 ng/mL was reached in AM-treated PC-3 cells compared to an IC_50_ value of >1000 ng/mL in PC-3 mock cells. Based on these results, AM was discussed as an enhancer of IL13 expression and sensitizer for IL13-PE therapy of PC [36]. In another study only low cytotoxicity with the targeted toxin IL13-PE38QQR was found in DU145 cells expressing IL-13R [37]. Transfection of the cells with the IL-13Rα2 subunit, which is an essential component for IL-13 binding and internalization [76], however, increased the affinity of the targeted toxin to IL-13 and enhanced its cytotoxicity and in vivo antitumor activity. Based on these data, sensitizing tumor cells for targeted toxin treatment by upregulation of IL-13 via gene transfer or by use of steroids or cytokines was discussed [37].

Taken together, there has to be a good balance between size and affinity of a targeted toxin and antigen expression on the tumor cells in order to achieve an optimal targeting.

#### 5.2.4. Reducing On-Target/Off-Tumor Toxicities

A main challenge, that is generally found in targeted therapies against cancer, is the presence of target antigens on normal cells which prevents the targeted toxins from acting in a tumor-specific manner. This means that targeted toxins might also damage such cells and that so-called on-target/off-tumor toxicities have to be expected in the clinic. For example, EGFR is widely distributed in human organs, including brain, heart, liver, skin, kidney, bone, breast, and lung [77]. Therefore, combination of anti-EGFR targeted toxins with other targeted agents agents or local activation by light might help to reduce on target/off-tumor side effects in future [33]. Besides the prostate, PSMA is expressed in the small intestine and in proximal tubules of normal kidneys and highest expression is found in the salivary glands [78,79]. Therefore, damage of these tissues could be a concern, when targeted toxins are used against PSMA. Indeed, xerostomia was found to be dose-limiting in anti-PSMA radioligand therapy. Interestingly, however, this coud be based on the preferential accumulation of the electronegatively charged radioligands in the salivary glands [80]. Xerostomia might, therefore, be radioligand specific and on-target/off-tumor toxicities in salivary glands might not be a feature of future targeted toxin therapies for PC.

## 6. Reduction of Immunogenicity

### 6.1. Challenges

Targeted toxins are artificial proteins that can be immunogenic in the patient, if they are not of human origin. Immunogenicity has to be given special consideration in the treatment of patients with PC, since their immune system is usually not compromised by disease or pretreatment. 

Studies with targeted toxins against hematological malignancies demonstrated that immunogenic reactions can occur already after one therapy cycle and leads to T-cell dependent and B-cell mediated formation of anti-drug antibodies (ADA) (rev. in [81]). ADAs can be either binding or neutralizing antibodies that alter the pharmacokinetics and reduce the efficacy of the targeted toxins. Moreover, they can induce adverse side effects like infusion reactions and hypersensitivity/anaphylactic reactions, which might be dose-limiting and result in discontinuation of treatment [82,83]. It is therefore necessary to reduce the immunogenicity of targeted toxins on both sides of the binding domains and of the toxin domains to increase patients’ safety. 

### 6.2. Solutions for Reduction of Immunogenicity

#### 6.2.1. Reducing the Immunogenicity of the Binding Domain

Many full-length monoclonal antibodies (mAbs) were generated via hybridoma techniques in mice. Therefore, targeted toxins containing such mAbs are expected to evoke immunogenic reactions in the patients [84]. Since many human anti-mouse antibodies react with the Fc region of the mAbs, the use of antibody fragments that do not have an Fc region, like scFv or dsFv, are preferred for the construction of targeted toxins with reduced immunogenicity. Immunogenicity of antibody fragments can be reduced by ‘humanization’, a genetic engineering process during that the complementarity determining regions (CDRs) are transferred to human variable region frameworks [85]. Moreover, humanized antibodies and antibody fragments can be directly generated by immunization of transgenic animals with introduced human immunoglobulin loci [86].

It is important that the specific antigen binding is not lost during the humanization process and that the cytotoxicity of targeted toxins, into which the humanized antibody or antibody fragments are incorporated, is still sufficient. In studies with PE-based targeted toxins containing murine or humanized anti-PSMA scFv as binding domains, specific binding to PSMA expressing LNCaP and C4-2 cells with comparable binding constants and cytotoxicity for both variants were described [18,44,46].

Another way to reduce the immunogenicity of the binding domain is to use human ligands. For the construction of targeted toxins against PC, e.g., the epidermal growth factor (EGF), the basic fibroblastic growth factor (bFGF), the interleukins-4 (IL-4) and -13 (IL-13), the lutenizing hormone-releasing hormone (LHRH), or human transferrin were used that are not expected to produce immunogenic reactions in the patients (Table 1). The corresponding targeted toxins showed comparable cytotoxicities with IC_50_ values in the low nM range against PC cells compared to those constructed with antibody fragments [31,34,36,37,38,39,41,56].

#### 6.2.2. Reducing the Immunogenicity of the Toxin Domain

Toxin domains elicit also immunogenic reactions in patients, when they originate from bacteria or plants. Besides the type I RIP protein Bouganin from *Bougainvillea spectabilis*, from which the de-immunized variant deBouganin exists [87], PE is a toxin, which was de-immunized by deletion of the translocation domain with exception of the furin clavage site [88] and by mutation of immunodominant B and T cell epitopes of domain III [89]. Targeted toxins containing the de-immunized PE domain showed reduced immunogenicity in preclinical and clinical trials [81]. Mutation of the B-cell epitopes, however, resulted in new T cell epitopes in the PE domain at the same time [89].

We have generated the targeted toxin EGF-PE24mut, consisting of the EGF ligand as binding and PE24mut as de-immunized toxin domain, and found a 6.1- to 11.9-fold reduced affinity (Kd = 26.6–36.9 nM) on LNCaP, C4-2, or PC-3 cells compared to the EGF ligand alone. However, it also had a 4–6.5-fold higher affinity compared to the targeted toxin EGF-PE40 containing the parental PE40 domain [31]. We presume that the smaller PE24mut domain resulted in less steric inhibition of the EGF ligand than the larger PE40 domain. The cytotoxicity of EGF-PE24mut on the PC cells, however, was about 11- to 120-fold lower than that of EGF-PE40. This can be traced back to the fact that the translocation domain of PE is deleted in parts in the PE24mut domain, so that its intracellular trafficking might be impaired. Our example shows that de-immunization of the toxin domain can lead to a higher affinity of a targeted toxin. On the other side, efficacy of targeted toxins can be diminished, e.g., when domains are affected by the de-immunization process, which are important for the function of the targeted toxin.

Minimizing the immunogenicity of the toxin domains can also be done by PEGylation [82,90]. PEG can mask immunogenic epitopes and is thought to be not immunogenic. However, in different studies anti-PEG antibodies have been detected in patients treated with PEGylated proteins, suggesting that PEG might be immunogenic itself and possibly enhance the immune responses to therapeutic agents [91].

The incorporation of human toxin domains into targeted toxins also helps to enhance safety, because they are not expected to be immunogenic in patients. In the last years, various targeted toxins against different tumor entities were generated containing the pro-apoptotic Bcl-2 family members Bax, Bak, or Bid/tBid, caspases, granzymes, endonucleases, RNases, or kinases (rev. in [92]). Meng and colleagues generated the targeted toxin immunocasp-3 consisting of the anti-PSMA scFv J591 as binding domain and human caspase-3 as toxin domain against PC. LNCaP cells that were co-incubated with immunocasp-3 secreting Jurkat cells were effectively killed. Moreover, tumor growth inhibition and enhanced survival was reached in mice with subcutaneous growing LNCaP tumors treated with lipofectamine-encapsuled immunocasp-3 or with the immunocasp-3 secreting Jurkat cells. Since human toxins lack a translocation domain with a cleavage site for cytosolic delivery, it was necessary to insert the furin-cleavable domain of *Diphteria toxin* between the two functional domains [45].

Targeted toxins containing human toxins are generally found to have lower cytotoxicity and antitumor activity compared to targeted toxins containing RIPs. The reason might be that human toxins can be inhibited by endogenous inhibitors in the cytosol. For example, caspase 3 can be inhibited by the X-linked inhibitor of apoptosis protein (XIAP) [93], RNaseA can be inhibited by ribonuclease inhibitors [94] and granzyme B can be blocked by the serin protease inhibitor B9 (serpin B9) [95].

Despite significant progress in the de-immunization of targeted toxins, there is still need for future research on how they can be administered in a truly safe manner in patients, including those with PC, without causing undesired immune responses.

## 7. Improvement of Intracellular Trafficking

### 7.1. Challenges

Prerequisite for a well-functioning targeted toxin is that the toxin domain is cleaved from the binding domain in the endosomes of target cells after internalization and that it reaches its site of action (ribosomes) in the cytosol. There is a risk of endosomal entrapment and lysosomal or proteasomal degradation of the toxin domain on its way through different cellular compartments (Figure 1). 

### 7.2. Solutions to Improve Intracellular Trafficking

To avoid degradation, agents can be added that cause a release of the toxin domains directly from the endosomes into the cytosol. These agents can be lysomotropic amines, carboxylic ionophores, or calcium channel antagonists, which temporarily weaken the membrane integrity, or cell penetrating peptides of different origin (rev. in [96]). An alternative is the use of glycosylated triterpenoids, like saponins, which are thought to associate with endo-/lysosomal membranes and mediate a pH value dependent delivery of the toxin domain into the cytosol [97].

In different studies with targeted toxin against PC, the Photochemical Internalization (PCI) technology was used to avoid lysosomal degradation and to enhance cytosolic release [29,33]. In PCI, photosensitizers that accumulate together with the targeted toxins in the endo-/lysosomes induce the formation of singlet oxygen after irradiation with light [98]. This leads to a damage of the endolysosomal membranes and release of the toxin domain into the cytosol. In a study of Bostad and colleagues, addition of the photosensitizer TPCS_2a_ (Amphinex) enhanced the cytosolic release of the targeted toxin IM7-saporin, consisting of an anti CD44 mAb as binding domain and saporin as toxin domain. As a result, the cytotoxic effects of the targeted toxin were enhanced in different CD44 expressing cancer cell lines, including the PC cell line DU145 [29]. In another study, synergistic cytotoxic effects were evoked in DU145 cells by PCI using the anti-EGFR targeted toxin cetuximab-saporin in combination with the photosensitizer TPPS_2a_ [33]. The use of PCI ensures that only cells, which are in the focus of the light source, are affected by the targeted toxin, thus avoiding on target/off-tumor side effects. The drawback of PCI, however, is that PC cannot be treated systemically with this method. The future scope of PCI-assisted targeted toxin therapy is therefore in the treatment of local prostate tumors or lymph nodes, which can be reached, e.g., via light emitting laser fibers.

## 8. Overcoming Apoptosis Resistance

### 8.1. Challenges

Resistance mechanisms against targeted toxins comprise a decrease in target antigen expression, an impaired intracellular trafficking, and apoptotic resistance in the tumor cells [99]. As mentioned above, a decrease in antigen expression could be overcome by increasing antigen expression via gene therapy or addition of drugs [36,37]. Intracellular trafficking of targeted toxins against PC can be enhanced by adding drugs or using PCI for enhanced cytosolic release [29,33].

Resistance against apoptosis, a typical hallmark of cancer, is based on an upregulation of anti-apoptotic Bcl-2 proteins (Bcl-2, Bcl-xl, Mcl-1) in PC cells [100] Interestingly, the pro-apoptotic members Bax and Bak are omnipresent in all tumor stages and mutations in these proteins that could affect their function are very rare [100,101]. Different combinatorial approaches with the aim of restoring sensitivity to apoptosis for effective target toxin therapy were therefore examined in PC cells [18,19,46,47].

### 8.2. Solutions to Overcome Apoptosis Resistance

The cytotoxicity of PE is based on the inhibition of protein biosynthesis of the target cells. Especially the expression of the anti-apoptotic proteins Mcl-1 and Bcl-2A1 is affected, because they have only short half-lives from a few minutes to a few hours due to constitutive protein turnover through poly-ubiquitination and proteasomal degradation [19]. We combined low doses of the anti-PSMA targeted toxin hD7(VL-VH)-PE40 with ABT-737, a BAD-like mimetic that inhibits the anti-apoptotic protein Bcl-2, Bcl-xl, and Bcl-w. The combination led to specific and synergistic cytotoxic effects on PSMA-expressing LNCaP and C4-2 cells and to a significantly prolonged survival of mice bearing C4-2 xenografts based on tumor growth inhibition [19]. Synergistic in vitro and in vivo effects were also found by combination of the targeted toxin D7(VL-VH)-PE40 with docetaxel chemotherapy [46]. Since docetaxel is known to sensitize PC cells for apoptosis by p53 activation and by altering the phosphorylation of Bcl-2 proteins, it is conceivable that it reduced the threshold for the induction of apoptosis by the targeted toxin. Baiz and colleagues generated a targeted toxin consisting of the anti-PSMA scFv J591 as binding and PE38QQR as toxin domain, called J591-PE. Combination of J591-PE with the pan-PI3K inhibitor ZSTK474 induced apoptosis in LNCaP and C4-2 cells. This was based on a reduction of Mcl-1 expression by the targeted toxin and on a dephosphorylation of BAD by the inhibitor that resulted in an enhanced inhibition of Bcl-2, Bcl-xl, and Bcl-w [47].

## 9. Conclusions

About three decades have passed since the generation of the first targeted toxins against cancer. Targeted toxins against hematological malignancies have already received clinical approval. Numerous others against solid tumors are currently still in early clinical trials, because it has taken a long time to develop them to the point where they can be tested safely for efficacy. Preclinical studies of targeted toxins against PC have shown that they have very high cytotoxic activity and that there are opportunities to overcome challenges to efficacy and safety. The use of targeted toxins in the context of a personalized medical approach that takes into account, for example, target antigen expression and possible resistance mechanisms in the tumor, could improve the treatment options for patients with advanced PC in the future.

## Figures and Tables

**Figure 1 biomedicines-09-00986-f001:**
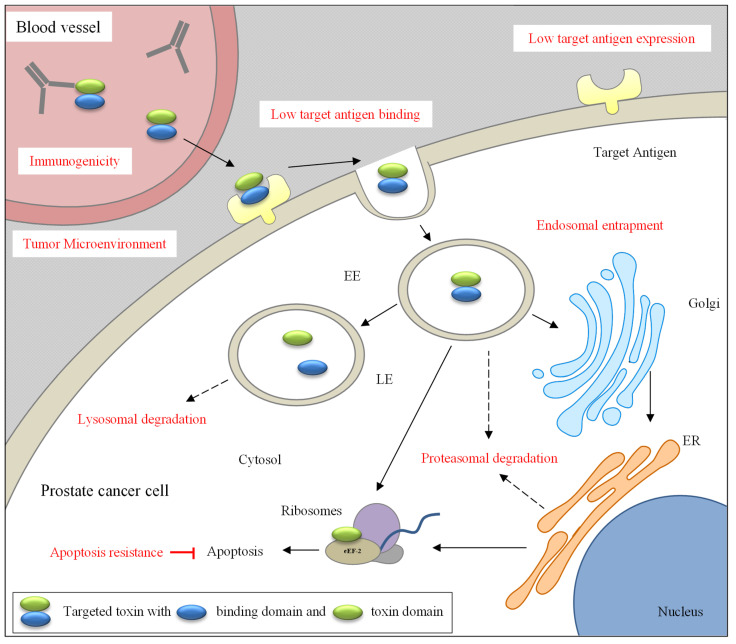
Intoxication of prostate cancer cells with targeted toxins and hurdles that might impair their efficacy. After systemic application targeted toxins are transported via the bloodstream to the tumor sites. They bind to antigens on the tumor cell surface and are internalized into early endosomes. After cleavage from the binding domain, the toxin domain is transported directly or via the Golgi network and the endoplasmatic recticulum into the cytosol. When RIP toxin domains are used, there is an inhibition of protein biosynthesis on the ribosomes with subsequent induction of apoptosis. Hurdles that might impair the efficacy of the targeted toxins are shown in red and comprise immunogenicity and low antigen binding of the targeted toxins, endosomal entrapment as well as lysosomal/proteasomal degradation during trafficking. Tumor cells can attenuate cytotoxicity by a dense tumor microenvironment, low antigen expression and apoptosis resistance. Abbreviations: EE, early endosomes; ER, endoplasmatic reticulum; LE, late endosomes.

**Figure 2 biomedicines-09-00986-f002:**
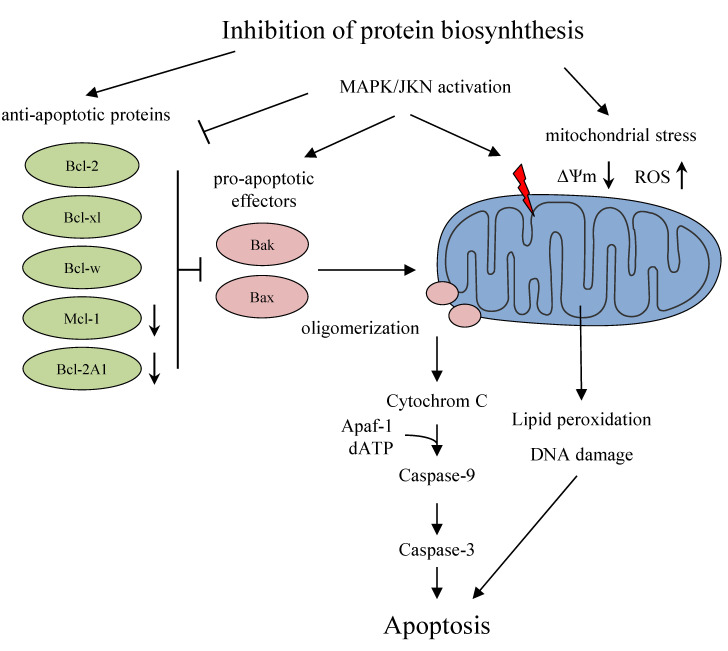
Proposed mechanisms by which targeted toxins induce apoptosis in cancer cells after inhibition of protein biosynthesis. ↑ upregulation; ↓, downregulation.

**Table 1 biomedicines-09-00986-t001:** Preclinical studies with targeted toxins against prostate cancer.

Antigen	Targeted Toxin	Binding Domain	Toxin Domain	Enhanced Efficacy/Safety by	Ref.
CD44	IM7-saporin	anti-CD44 mAb (clone IM7)	Saporin	combination with PCI	[29]
CDPD1	ch25A11-Sap	anti-CDCP1 mAb 25A11	Saporin		[30]
EGFR	EGF-PE40 EGF-PE24mut	EGF	PE40PE24mut	human binding domain, de-immunized toxin domain	[31]
	scFv2112-ETA’ (from cetuximab) scFv1711-ETA’ (from panitumumab)	anti-EGFR scFv	ETA‘		[32]
	cetuximab-saporin	anti-EGFR mAb cetuximab	Saporin	combination with PCI	[33]
FGF	bFGF-SAP	bFGF	saporin		[34]
Her2	scFv(FRP5)-ETA	anti-HER2 scFv	ETA		[35]
IL13R	IL-13PE	human IL-13	PE38	human binding domain, enhancing target antigen expression, intratumoral injection	[36]
	IL13-PE38QQR	human IL-13	PE38QQR	human binding domain, enhancing target antigen expression	[37]
	IL13-PE38QQR	human IL-13	PE38QQR	human binding domain	[38]
IL4R	IL4-CTx	human IL-4	PE	human binding domain, intratumoral injection	[39]
	hIL4-PE4E	human IL-4	PE mutant	human binding domain	[40]
LHRH	LHRH-RNase A conjugate	LHRH	bovine RNaseA	human binding domain	[41]
p185 erbB-2	AR209	anti-p185^erbB-2^ scFv e23Fv	PE38KDEL		[42]
PSMA	JVM-PE24X7	anti-PSMA sd Ab	PE24X7	de-immunized toxin domain	[43]
	hD7-1(VL-VH)-PE40	anti-PSMA scFv	PE40	combination with ABT-737	[19]
	hD7-1(VL-VH)-PE40 hD7-1(VL-VH)-PE24 hD7-1(VL-VH)-PE24mut	humanized anti- PSMA scFv	PE40 PE24 PE24mut	de-immunized toxin domains	[44]
	D7(VL-VH)-PE40	humanized anti-PSMA scFv	PE40	combination with ABT-737	[18]
	immunocasp-3	anti-PSMA scFv J591	rev caspase-3	human binding and toxin domain	[45]
	hD7-1(VL-VH)-PE40	anti-PSMA scFv	PE40	combination with docetaxel	[46]
	J591PE	anti-PSMA scFv (J591)	PE38QQR	combination with pan-PI3K inhibitor	[47]
	A-dmDT390-scfbDb(PSMA)	anti-PSMA sdAb J591	truncated diphtheria toxin (DT)		[48]
	D7-VH(Yol)VL-PE40 D7-VH(GS)VL-PE40 His D7-VH(GS)VL-PE40 D7-VL(GS)VH-PE40 His-D7-VL(GS)VH-PE40	anti-PSMA scFv	PE40	enhanced affinity by changing scFv domain orientation	[49]
	hJ591-SAZAP	hJ591	saporin	humanized mAb as binding domain	[50]
	D7-PE40	anti-PSMA scFv D7	PE40		[51]
	A5-PE40	anti-PSMA scFv A5	PE40		[52]
	A5-PE40	anti-PSMA scFv A5	PE40		[53]
	E6-dgA	anti-PSMA mAb E6	ricin A chain		[54]
	J591-smpt-nRTA	anti-PSMA mAbs J591 PEQ226.5 PM2P079.1	RTA, native orrecombinant		[55]
Tf	Tf-SapTf-A_RCA_	transferrin	Saporin or Ricin A	human binding domain, combination with monensin and chloroquine	[56]

Abbreviations: bFGF, basic fibroblast growth factor; CDCP1, UB domain-containing protein 1; IL13, interleukin 13; LHRH, luteinizing hormone-releasing hormone; PCI, photochemical internalization; PE, ETA′, Pseudomonas Exotoxin A; RTA, Ricin A chain; scFv, single chain variable fragment; sdAb, single domain antibody.

**Table 2 biomedicines-09-00986-t002:** Strategies to optimize the efficacy of targeted toxins against prostate cancer.

	Challenges	Solutions	Ref.
**Effective Targeting**			
*Surmounting the TME*	TME prevents extravasation and tumor penetration of targeted toxins	Surmounting the TME by - intratumoral injection of the targeted toxins - pre-damage of tumor masses	[36,39,46]
*Enhancing tumor penetration and affinity*	Large size of targeted toxins prevents tumor pentration	Reducing the size of the binding domain Reducing the size of the toxin domain	[31,44]
	Targeted toxins have low binding affinity	Enhancing affinity by changing the arrangement of the functional domains of a targeted toxin	[49]
*Enhancing target antigen expression*	Low target antigen expression on the PC cells	Enhancing target antigen expression by - gene transfer - drugs	[36,37]
*Reducing on-target/off-tumor toxicities*	Targeted toxins might harm normal cells that express the target antigen	Reducing on-target/off-tumor toxicity by - intratumoral injection of the targeted toxins - local activation of the targeted toxins	[29,33,36,39]
**Reduction of immunogenicity**			
*Reducing the immunogenicity of the binding domain*	Targeted toxins with non-human binding domain are immunogenic in PC patients	Reducing immunogenicity by - humanization of antibody fragments - use of human ligands	[18,34,36,37,38,39,40,44,50,56]
*Reducing the immunogenicity of the toxin domain*	Targeted toxins with non-human toxin domain are immunogenic in PC patients	Reducing immunogenicity by - de-immunization - use of human toxins	[31,41,44,45]
**Improvement of intracellular trafficking**	lysosomal and proteasomal degradation of the targeted toxins	Enhancing cytosolic release by - addition of drugs - photochemical internalization	[29,33]
**Overcoming apoptotic resistance**	apoptosis resistance of PC cells	Enhancing sensitivity to apoptosis by combination with - BH3 mimetics - chemotherapy - kinase inhibitors	[18,19,46,47]

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
