# Peer review of "Targeted Toxins for the Treatment of Prostate Cancer"

_biomedicines, 2021, doi:10.3390/biomedicines9080986_

Round 1

Reviewer 1 Report

Please identify the attached file.

Author Response

We thank reviewer # 1 for the helpful comments that improve the content of the manuscript. The manuscript was revised accordingly and changes are shown in red. Points of criticism were answered point by point as follows:

  1. Abstract of the manuscript is very brief. Author is directed to rewrite the abstract in a readers-friendly manner, which could reflect the whole manuscript in real way.

The abstract was rewritten accordingly.

  1. Line 54: “After application they are transported via the bloodstream to the tumor and bind with help of their binding domain to antigens on the surface of the tumor cells”. Please address the following points:
  2. i) What does “after application” mean? Please elaborate the route of administration of targeted toxins, for example, oral, parental, topical, rectal, vaginal, vector-directed etc. Please support the statement with relevant references.

Line 76: The route of administration was specified to “systemic” application. The reference of Pak et al., 2014 was added.

  1. ii) Author has not mentioned how do the targeted toxins specifically bind with the tumor cells.

They may bind to normal cells as well. Author is advised to add explanation about specific/nonspecific binding of targeted toxins to the cells.

Lines 68-74: More explanation about exchanging the natural binding domains of RIPs against specific binding domains was added.

Lines 281-296: A new sub-chapter “Reducing on-target/off-tumor toxicities” was added.

  1. Line 57: “In the endosomes, the toxin domain is cleaved from the binding domain and there are unfolding steps that lead to conformational changes of the toxin domains to travel through the cellular compartments into the cytosol”.

Please address the following points:

  1. i) Please add the detailed mechanism(s) of cleavage of toxin domain. Additionally, please provide a quality elaborative figure.
  2. ii) What are the unfolding steps? Author should provide details of these steps. It would be better if author add a high-quality figure which could explain all these steps one by one.

Fig. 1 gives an overview of the intoxication steps. Since the review focuses on targeted toxins against prostate cancer and the cleavage and unfolding mechanisms, that are individually and toxin-dependent, would go beyond the scope, a detailed illustration about cleavage has not been made. Instead, more information about cleavage and unfolding of the targeted toxins was added to the text (lines 78-86). The references of Michalska et al., 2015 and of Blum et al., 1991 were added, where the reader can find the detailed cleavage mechanisms.

  1. Line 68-69: How would inhibition of protein biosynthesis lead to induce intrinsic apoptotic pathway and death of tumor cells? Author is advised to add intensive explanation and provide the mechanism(s) along with high-quality elaborative figure(s).

Information explaining the links between inhibition of protein biosynthesis and induction of apoptosis was added in the text (lines 108-117) Moreover, an overview is now given in Figure 2.

  1. Line 71-73: Please support this statement with relevant studies.

Now line 90: The reference of Nowakowska-Gołacka et al., 2019 was added.

  1. Lines 86-91: Please add relevant references.

Now lines 130-134: The references of Piascik, 1999, Fancher et al., 2019, and Jen et al., 2020, were added.

  1. Lines 108-110: “In the following sections challenges in the generation of targeted toxins against PC in view of effective targeting, reduction of immunogenicity, improvement of intracellular trafficking and overcoming resistance are discussed”. After this statement, author has discussed effective targeting, reduction of immunogenicity, improvement of intracellular trafficking and overcoming resistance in detail. Here, I have following suggestion to the author:
  2. i) The section “Surmounting the TME” should be further subdivided into following subheadings: (a) challenges (b) solution/strategy to overcome these challenges.
  3. ii) Similarly, the section “Enhancing tumor penetration and affinity” should be subdivided into: (a) challenges (b) solution/strategy to overcome these challenges.

iii) Likewise, all other headings and subheading should be divided in the directed manner.

  1. iv) Finally, all the challenges and their solutions/overcoming strategies must be presented in tabulated form.

The sections were further divided into a) challenges and b) solutions to overcome these challenges. Challenges were also added in Fig. 1. Table 2 was added according to the reviewer’s recommendations.

  1. Wherever applicable, please provide the specific name and type of prostate cancer cell lines. For example, refer to lines 198, 199, 235, 256, 260, 331, 335, 343, 346, 351. Writing only “PC cells” is not sufficient.

The specific names of the PC cells were added (now lines 217, 256, 272, 329, 349, 440, 441, 449)

  1. Certain references cited in the manuscript are very old, such as Ref# 23 (1996), 27 (1997), 29 (1994), 30, 31 (1999), 47 (1998), 49 (1989), 53 (1999), 61 (1991), 62 (1998), 72 (1995), 75 (1993), 89 (1996). Author is advised to cite recent relevant studies (not older than 5-10 years).

We agree with the reviewer that the references have to be up-to-date. References #23, 27, 29, 30, 31, and 47 contain original studies about targeted toxins against PC. Since we want to give a comprehensive overview, we decided to keep these studies in the article.

Reference #49 was replaced by the reference of Belli et al., 2018.

To our knowledge, reference #53 is the only one, which demonstrated decompression of blood vessels by taxanes and was therefore retained.

Reference #61 was exchanged against the reference of Zhou et al., 2012.

References #62, 73 and 75 were deleted.

Reference #89: The expression of Bcl-2 family proteins in human prostate tumors was systematically examined in the mid to late 90s. Therefore this reference was retained.

  1. Conclusion of the manuscript is not significant. Author is advised to rewrite this section in few lines with scientific approach, taking the key points from the contents of the manuscript.

The discussion was rewritten (page 17).

  1. Author is advised to provide a clear and well demonstrated graphical abstract for the manuscript.

Figure 1 represents the content of the review and can be used as graphical abstract.

Reviewer 2 Report

This is a very nice narrative review written by a key opinion leader in the field of targeted toxins.

  • The review cites lot of previous research work form the author, and there is no mention about where and how the bibliography was selected. Did the authors used any specific groups of terms? Which data base were consulted?
  • It would be really interesting also to contextualize and explore deeply through the bibliography the putative treatment application in a focal intraprostatic manner. There are already early studies exploring similar approaches.
  • Also to deeply explore the possible adverse effect with the mentioned targets (the epidermal growth factor receptor (EGFR) and the prostate specific membrane antigen (PSMA)) as they are not exclusive from the prostate.

Author Response

Reviewer 2

We thank reviewer # 1 for the helpful comments that improve the content of the manuscript. The manuscript was revised accordingly and changes are shown in red. Points of criticism were answered point by point as follows:

Comments and Suggestions for Authors

This is a very nice narrative review written by a key opinion leader in the field of targeted toxins.

The review cites lot of previous research work form the author, and there is no mention about where and how the bibliography was selected. Did the authors used any specific groups of terms? Which data base were consulted?

Information about database search was added (lines 142-143).

It would be really interesting also to contextualize and explore deeply through the bibliography the putative treatment application in a focal intraprostatic manner. There are already early studies exploring similar approaches.

The background of focal therapy was discussed in more detail (lines 194-211). The references of van den Bos et al., 2014 and of Rogers et al., 2021 were added.

Also to deeply explore the possible adverse effect with the mentioned targets (the epidermal growth factor receptor (EGFR) and the prostate specific membrane antigen (PSMA)) as they are not exclusive from the prostate.

Extraprostatic EGFR and PSMA expression and possible on target/off-tumor effects during treatment  with targeted toxins is now discussed in the new chapter “Reducing on-target/off-tumor toxicities”  (page 12 ).

Round 2

Reviewer 1 Report

Author has tried his best to address all the comments. The manuscript has been sufficiently improved.

Reviewer 2 Report

Thank you for the review and the revision version.

Congratulations